# Coordinating virus research: The Virus Infectious Disease Ontology

**John Beverley** [1,2]*, **Shane Babcock**[2,3], **Gustavo Carvalho**[4], **Lindsay G. Cowell**[5], **Sebastian Duesing**[6], **Yongqun He**[7], **Regina Hurley**[2,8], **Eric Merrell**[1,2], **Richard H. Scheuermann**[9,10,11], **Barry Smith**[1,2]

**1** Department of Philosophy, University at Buffalo, Buffalo, NY, United States of America, **2** National Center for Ontological Research, Buffalo, NY, United States of America, **3** Air Force Research Laboratory, Wright Patterson Air Force Base, Riverside, OH, United States of America, **4** Department of Cognitive Science, Northwestern University, Evanston, IL, United States of America, **5** Department of Clinical Sciences, University of Texas Southwestern Medical Center, Dallas, TX, United States of America, **6** Department of Philosophy, Loyola University, Chicago, IL, United States of America, **7** Computational Medicine and Bioinformatics, University of Michigan Medical School, He Group, Ann Arbor, MI, United States of America, **8** Department of Philosophy, Northwestern University, Evanston, IL, United States of America, **9** Department of Informatics, J. Craig Venter Institute, La Jolla, CA, United States of America, **10** Department of Pathology, University of California, San Diego, CA, United States of America, **11** Division of Vaccine Discovery, La Jolla Institute for Immunology, La Jolla, CA, United States of America

* johnbeve@buffalo.edu

**Data Availability Statement:** The Virus Infectious Disease Ontology artifact can be found in the following Github repository: https://github.com/infectious-disease-ontology-extensions/ido-virus. The Coronavirus Infectious Disease Ontology

## Abstract

The COVID-19 pandemic prompted immense work on the investigation of the SARS-CoV-2 virus. Rapid, accurate, and consistent interpretation of generated data is thereby of fundamental concern. Ontologies–structured, controlled, vocabularies–are designed to support consistency of interpretation, and thereby to prevent the development of data silos. This paper describes how ontologies are serving this purpose in the COVID-19 research domain, by following principles of the Open Biological and Biomedical Ontology (OBO) Foundry and by reusing existing ontologies such as the Infectious Disease Ontology (IDO) Core, which provides terminological content common to investigations of all infectious diseases. We report here on the development of an IDO extension, the Virus Infectious Disease Ontology (VIDO), a reference ontology covering viral infectious diseases. We motivate term and definition choices, showcase reuse of terms from existing OBO ontologies, illustrate how ontological decisions were motivated by relevant life science research, and connect VIDO to the Coronavirus Infectious Disease Ontology (CIDO). We next use terms from these ontologies to annotate selections from life science research on SARS-CoV-2, highlighting how ontologies employing a common upper-level vocabulary may be seamlessly interwoven. Finally, we outline future work, including bacteria and fungus infectious disease reference ontologies currently under development, then cite uses of VIDO and CIDO in host-pathogen data analytics, electronic health record annotation, and ontology conflict-resolution projects.

artifact can be found in the following Github repository: https://github.com/CIDO-ontology/cido.

**Funding:** Sources of funding for this article for John Beverley and Shane Babcock stem from the NIH / NLM T5 Biomedical Informatics and Data Science Research Training Programs. Barry Smith's source of funding stemmed from the NIH under NCATS 1UL1TR001412 (Buffalo Clinical and Translational Research Center). No other co-authors were funded to pursue work on this project. Moreover, the funders had no role in the study design, data collection and analysis, decision to publish, or preparation of the manuscript.

**Competing interests:** The authors have declared that no competing interests exist.

## Introduction

The value of cross-discipline meta-data analysis has been evident in the COVID-19 pandemic. Early in the pandemic, for example, prostate oncologists [1, 2] attempted to leverage existing research on enzymes crucial in host cell penetration by SARS-CoV-2 to explain differences in disease severity across sex [3, 4]; immunologists combined insights from research on SARS-CoV-1 and MERS-CoV with chemical compound profiles to identify treatment options for SARS-CoV-2 [5–7]; pediatric researchers, observing that children have fewer nasal epithelia susceptible to SARS-CoV-2 infection than adults, suggested this difference may explain symptom disparities between the two groups [8, 9]. The sheer volume of data collected by life-science researchers, the speed at which it is generated, range of its sources, quality, accuracy, and urgency of need for assessment of usefulness, has resulted in complex, multidimensional datasets, often annotated using discipline- or institution-specific terminologies and coding systems that lead to data silos [10–12].

Data silos emerge in life science research when data concerning an area of research is stored in a manner that makes it accessible to one group, but inaccessible to others. The use of proprietary information systems, differing storage methods, and distinct coding standards across life science that is characteristic of such silos undermines interoperability, meta-data analysis, pattern identification, and discovery across disciplines [13, 14]. Ontologies–interoperable, logically well-defined, controlled vocabularies representing common entities and relations across disciplines using consensus terminologies–constitute a well-known solution to these problems through mitigation of the formation of data silos. The need for rapid analysis of evolving datasets representing coronavirus research motivated the development of the Virus Infectious Disease Ontology (VIDO; https://bioportal.bioontology.org/ontologies/VIDO), comprised of textual definitions for terms and relations and logical axioms supporting automated consistency checking, querying over datasets, and interoperability with other ontologies. VIDO is an extension of the widely-used Infectious Disease Ontology Core (IDO Core; https://bioportal.bioontology.org/ontologies/IDO) [15, 16], which comprises terminological content common to all investigations of infectious disease. VIDO extends IDO with terms specific to the domain of infectious diseases caused by viruses and provides a foundation for ontologies representing specific viral infectious diseases, such as COVID-19.

VIDO is available under the Creative Commons Attribution 4.0 license (https://creativecommons.org/licenses/by/4.0/) and its current and past versions can be found at the National Center for Biomedical Ontology (NCBO) Bioportal [17], the Ontobee repository (http://www.ontobee.org/), and the Ontology Lookup Service (https://www.ebi.ac.uk/ols/index). VIDO was developed in collaboration with relevant domain experts, including immunologists and virologists, and by drawing on the expertise of the IDO developers to ensure alignment with principles outlined by the Open Biological and Biomedical Ontology (OBO) Foundry [18], thereby supporting interoperability with existing Foundry ontologies [19]. VIDO development is a transparent process, with all discussions available on GitHub (https://github.com/infectious-disease-ontology-extensions). All aspects of development, including addition of new terms, are driven by the needs of researchers investigating viruses and nearby domains. The ontology is thus not viewed as exhaustive of the domain of virus research but remains sensitive to evolving knowledge.

## Methods

### OWL, Protégé, Mace4, and Prover9

VIDO is formally represented in the OWL2 Web Ontology Language (https://www.w3.org/TR/owl2-overview/). OWL2 is an expansion of the Resource Description Framework (RDF; https://www.w3.org/TR/rdf-primer/) and of RDF Schema, which represent data as sets of subject-predicate-object directed graphs, and which can be queried using the SPARQL Protocol and RDF Query Language (https://www.w3.org/TR/sparql11-query/). OWL2 supplements these languages by allowing for description of classes, members of classes, relations among individuals, and annotation properties. Formally, the OWL2 vocabulary can be mapped to a decidable fragment of first-order logic, meaning there is an algorithm which can determine the truth-value for any statement expressed in the language in a finite number of steps [20]. Restricting expressions to a decidable language allows automated consistency and satisfiability checking [21]. VIDO was developed using the Protégé-OWL editor (https://protege.stanford.edu/) and tested against OWL reasoners such as HermiT [22] and Pellet [23]. Additionally, logical axioms underwriting these ontologies were translated into the Common Logic Interchange Format, and subsequently evaluated using the Mace4 model checker and Prover9 proof generator within the Macleod toolkit (https://github.com/thahmann/macleod).

### Alignment with OBO Foundry ontologies

Ontologies are widely used in bioinformatics, supporting data standardization, integration, sharing, reproducibility, and automated reasoning. The Gene Ontology (GO; https://bioportal.bioontology.org/ontologies/GO), for example, maintains species-neutral annotations of gene products and functions, and since its inception in 1998 it has inspired an explosion of biomedical ontologies covering all domains of the life sciences [19, 24, 25]. These early developments led to worries, however, that data silos–the very problem ontologies were designed to address–might reemerge [10] as researchers developed ontologies using concepts local to their discipline. By 2007, the Open Biomedical and Biological Ontologies (OBO) Foundry [18] was created to provide guidance for ontology developers and promote alignment and interoperability. OBO Foundry design principles require that ontologies: use a well-specified syntax that is unambiguous, with a common space of identifiers; that they be openly available in the public domain, have a specified scope, be developed in a modular fashion in a collaboration with ontologists covering nearby domains, and import a common set of relations from the Relations Ontology (RO; https://obofoundry.org/ontology/ro.html). The OBO library (http://obofoundry.org/) presently consists of over 250 ontologies, including some externally developed ontologies such as the NCI Thesaurus (https://ncithesaurus.nci.nih.gov/ncitbrowser/) and the NCBI Taxonomy (https://www.ncbi.nlm.nih.gov/taxonomy). It also contains some constructed *ab initio* to satisfy OBO principles. At its core is Basic Formal Ontology (BFO; https://bioportal.bioontology.org/ontologies/BFO), a top-level ontology covering general classes such as *material entity*, *quality*, *process*, *function* and *role* [10, 26–29] which provides the architecture "on which OBO Foundry ontologies are built." BFO is, moreover, an ISO/IEC approved standard 21838–2 (https://www.iso.org/standard/74572.html).

Where BFO is domain-neutral, other OBO Foundry ontologies represent types of entities in more specific domains, using terms such as *disease*, *cell division*, *surgical procedure*, and so forth. Ideally, domain ontologies are constructed using a methodology for formulating definitions through a process of downward population from BFO. The resulting alignment with BFO, and the conformance to OBO Foundry principles, foster integration across ontologies. VIDO was designed with alignment and conformance in mind. Development of each ontology

follows metadata conventions adopted by many OBO Foundry ontologies [30]. These conventions require that every term introduced into the ontology has a unique IRI, textual definitions, definition source, designation of term editor(s), and preferred term label. In the interest of coordinating development with existing OBO ontologies, VIDO developers imported terms where possible from existing OBO library ontologies and constructed logical definitions using imported terms. Development was guided by best practices for definition construction [10, 31]. New primitive terms were introduced when needed after consultation with domain experts, review of relevant literature, and careful examination of the OBO library to avoid redundancy.

## Hub and spokes approach

VIDO follows the "hub and spokes" methodology [32, 33] for ontology development. That is, VIDO is a spoke ontology, extending from the Infectious Disease Ontology Core (IDO Core; https://bioportal.bioontology.org/ontologies/IDO) as its hub. IDO Core is an OBO ontology consisting of terms, relations, natural language definitions and associated logical axioms representing phenomena common across research in infectious diseases [15]. IDO Core has long provided a base from which more specific infectious disease ontologies extend, and it has been recently updated to keep pace with scientific and top-level architecture changes [16]. Extensions of IDO Core covering specific infectious diseases are created, first, by importing needed terms from IDO Core and other OBO Foundry ontologies, and second, by constructing the domain-specific terms where needed to adequately characterize entities in the relevant domain. Fig 1 illustrates example extensions, such as the Brucellosis Infectious Disease Ontology (IDOBRU; https://bioportal.bioontology.org/ontologies/IDOBRU) the Influenza Infectious Disease Ontology (IDOFLU; https://bioportal.bioontology.org/ontologies/FLU), and more recently the Coronavirus Infectious Disease Ontology (CIDO; https://bioportal.bioontology.org/ontologies/CIDO). Each aims to be semantically interoperable with OBO library ontologies [11, 34–36]. VIDO was designed to occupy the ontological space between such virus-specific ontologies and IDO Core. As a result, more specific virus-related ontologies such as CIDO [37] and IDOFLU are being curated to extend directly from VIDO, rather than directly from IDO Core.

## Results

### The Virus Infectious Disease Ontology

VIDO takes IDO Core as its starting point, but also imports terms relevant to the domain of viruses from several other OBO Foundry ontologies, such as GO, the Ontology for General Medical Science (OGMS; https://bioportal.bioontology.org/ontologies/OGMS) and the Ontology for Biomedical Investigation (OBI; https://bioportal.bioontology.org/ontologies/OBI) [14]. The color- coded Fig 1 illustrates several importing relationships which provide the basis for VIDO definitions which we examine in what follows.

**Acellular structure.** Like IDO Core, VIDO imports from OGMS. Examples of such imported terms are:

*disorder* = def Material entity that is a clinically abnormal part of an extended organism.

A part of a material entity is "clinically abnormal" if it is not expected in the life plan for entities of the relevant type and is causally linked to elevated risk–that is, risk exceeding some threshold–of illness, death, or disfunction [38]. *Extended organism* is imported from OGMS and *organism* from OBI, where they are defined as follows:

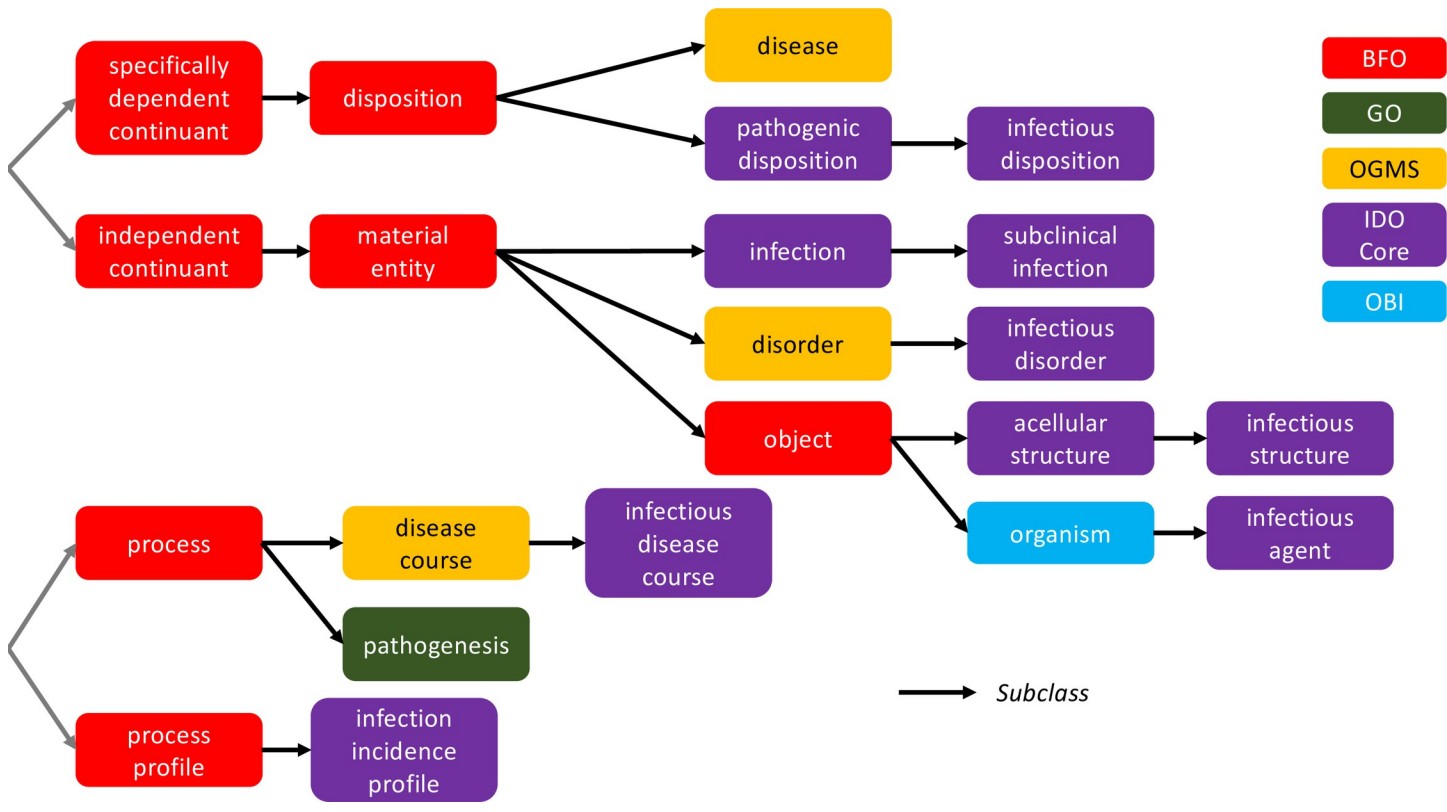

**Fig 1. Relationships among BFO, GO, OGMS, IDO Core, and OBI.**

*extended organism* = def An object aggregate consisting of an organism and all material entities located within the organism, overlapping the organism, or occupying sites formed in part by the organism.

*organism* = def A material entity that is an individual living system, such as animal, plant, bacteria, or virus, that is capable of replicating or reproducing, growth and maintenance in the right environment. An organism may be unicellular or made up, like humans, of many billions of cells divided into specialized tissues and organs.

Here we run into the first of several ontological puzzles that emerged while developing VIDO. On the one hand, this definition aligns with common usage of the term "organism" among researchers for whom its instances are *cellular entities* [39, 40]. On the other hand, the textual definition includes viruses among its instances, which are in every case acellular. Debates over *organism* (https://github.com/OBOFoundry/COB/issues/6) among ontology developers have resulted in deprecation of the OBI term in favor of a nearby term from the Common Anatomy Reference Ontology (https://bioportal.bioontology.org/ontologies/CARO) with the label: *organism or virus or viroid.* At first glance, this appears to avoid the preceding worries, but further inspection reveals that this class is annotated as being an "exact synonym" of organism, and so suffers from the same issues raised above. Even if we put aside this latter issue, however, there are still two further concerns. First, introducing disjunctive classes is *ad hoc* [10]. Second, this disjunctive class leads naturally to debates over whether viruses are alive, since it classifies viruses alongside paradigmatic living entities. Decades of discussion have not resolved this question [41–46], and it is not obvious that we need an answer for the purposes of ontological modelling. It is unclear where consensus will land; in the interest of future-

proofing our ontologies we should provide virus content in a way that neutral in regard to this issue to the maximal extent that this is possible. Rather than introduce an *ad hoc* disjunctive class, IDO Core, VIDO, and CIDO developers collaborated to add the following classes to IDO Core [47]:

*self-replicating organic structure* = def Object consisting of an organic structure that is able to initiate replication of its structure in a host.

*acellular self-replicating organic structure* = def Self-replicating organic structure comprised of acellular organic parts.

Which is imported to VIDO as the parent class of the term *virus*. The term *virus* is imported from the NCBITaxon [48] (https://bioportal.bioontology.org/ontologies/NCBITAXON), alongside terms relevant to virology, such as *prion* and *satellite*.

**Virus.** The NCBITaxon provides an extensive list of life science terms, but it has its limitations. As discussed in [16], NCBITaxon categorizes virus terms using the International Committee on Taxonomy of Viruses (ICTV). While an impressive taxonomy, the ICTV exhibits gaps in virus classification [49, 50]. Additionally, NCBITaxon combined with standard ontology engineering tools such as Ontofox (http://ontofox.hegroup.org/) [51] often leads to ontology developers importing superfluous portions of ICTV structured hierarchies, resulting in overwhelming taxonomies that are challenging for users to navigate. Fig 2 illustrates such a taxonomy found in IDOBRU, but importing an entire ICTV hierarchy is not uncommon (see for example the Schistosomiasis Ontology (IDOSCHISTO) [16]). Lastly, the NCBITaxon does not provide textual definitions for most terms within its scope. As stated, we seek to respect OBO Foundry metadata conventions [30] and ontology engineering best practices [10, 31]; consequently, *virus* and other terminological content in VIDO must have textual definitions supplied.

Standard definitions of "virus" provide a starting point for a textual definition, but caution is once again needed. Viruses are often described as obligate pathogens [52, 53], since virus replication requires host machinery for production and assembly of viral components. However, defining a class *virus* solely in terms of what viruses typically do runs the risk of overlooking what viruses are, materially speaking. Compare: *Homo sapiens* are obligate aerobes, but this is no definition of the class. Insofar as we are defining the virus class, it is better to attend to genetic and structural components common to all viruses, and best to define the material entity in a way that captures obligate pathogenicity. VIDO accordingly defines:

*virus* = def Acellular self-replicating organic structure with RNA or DNA genetic material which uses host metabolic resources for RNA or DNA replication.

VIDO developers have contacted NCBITaxon developers proposing that the definitions we provide below be added to respective NCBITaxon terms. Other requests have been submitted, for example, to update the label of the NCBITaxon term from "Viruses" to "virus" in order to avoid ambiguous reference between classes and their instances [10].

Rather than import in accordance with the ICTV taxonomy, subclasses of *virus* are imported from NCBITaxon in alignment with the Baltimore Classification [54], which groups viruses into seven exhaustive classes based on genetic structure. For example, one subclass of *virus* is:

*positive-sense single-stranded RNA virus* = def Virus with genetic material encoded in single-stranded RNA that can be translated directly into proteins.

Which is a class representing one of the seven Baltimore Classification categories. These Baltimore Classification classes provide parent classes from which terms for more specific viruses can extend. Fig 3 illustrates how the Baltimore Classification appears in the Protégé visualization of the class *positive-sense single-stranded RNA virus* alongside viral replication pathways underwritten by genetic differences in viruses.

**Fig 2. NCBITaxon imported to IDOBRU, located on Bioportal.**

By incorporating the Baltimore Classification, we provide developers of virus-specific domain ontologies an ontological representation of viruses that is simpler and easier to navigate than the ICTV, which currently underwrites the NCBITaxon.

VIDO subclasses of *virus* include those common in virology research, such as *bacteriophage*–viruses that infect bacteria–*virophage*–viruses that infect viruses–*oncovirus*–viruses that cause cancer–and *mycovirus*–viruses that infect fungi. As well as:

*virion* = def Virus that is in its assembled state consisting of genomic material (DNA or RNA) surrounded by coating molecules.

Some researchers use "virion" and "virus" synonymously [55]. Some define "virion" so that instances only exist outside host cells [56], or they distinguish virions outside host cells from those inside host cells, calling the former "mature virions." Some claim "virion" is best understood as analogous to a sperm cell [57, 58]. Ontologically speaking, one might model the relationship between a virus and its virion in a variety of ways: virion is to a virus as human infant is to human, or as human student is to human, or as human gamete is to human. Treating virions as akin to gametes is uncommon among researchers. Between the remaining options, we adopt the first, treating *virion* as a type of *virus*, since adopting the alternative would suggest a virion is simply a virus that is in a specific context, a result that overlooks the importance of genomic assembly to identifying virions.

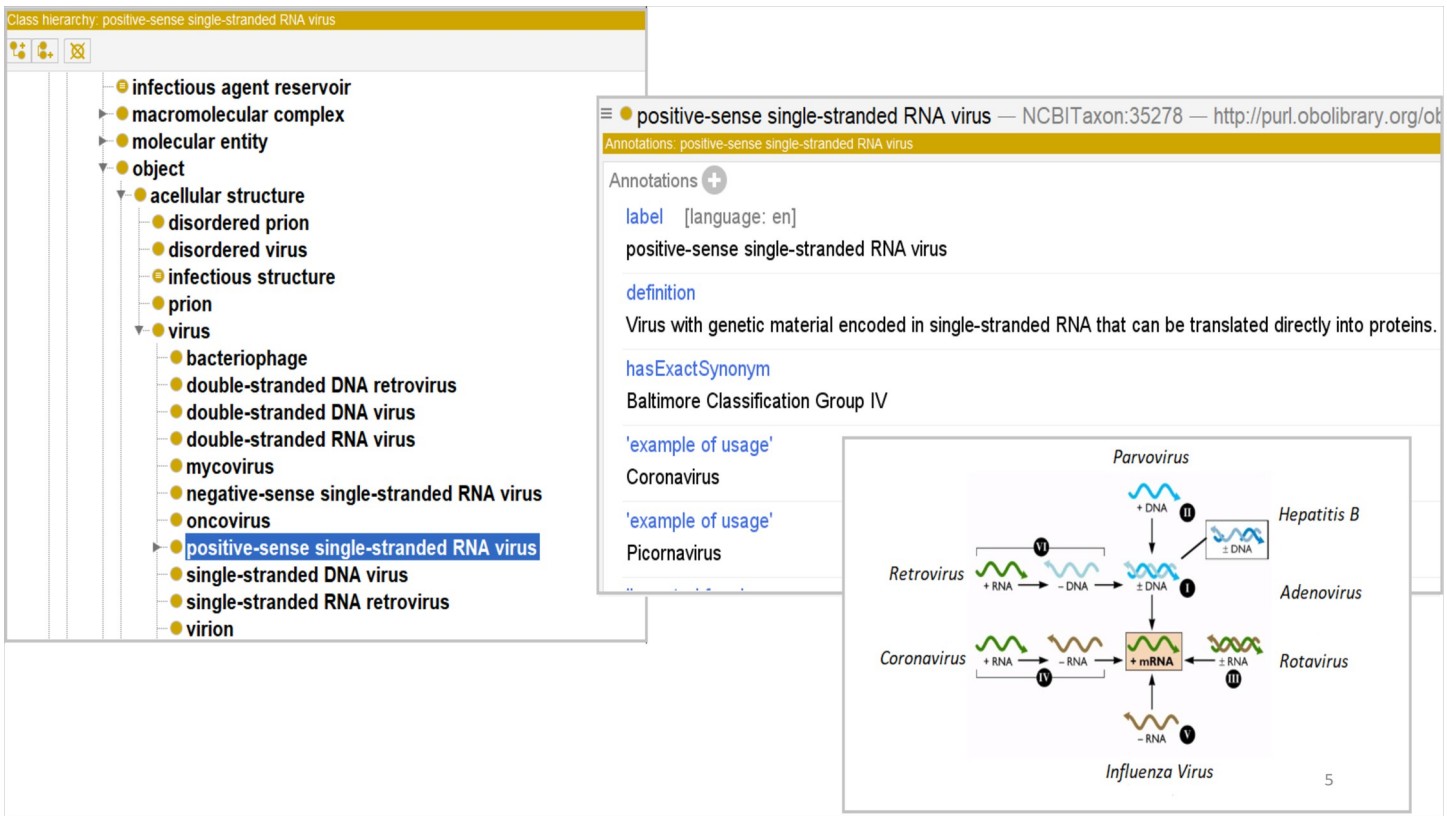

**Fig 3. VIDO representation of the Baltimore Classification in Protégé Editor.**

Incidentally, some viruses do not replicate faithfully, perhaps resulting in genetically distinct mutants or–in extreme cases–an inactive aggregate of virion components. Virus mutations may undermine host immune system recognition of viral threats, as evidenced by the difficulty in developing vaccines for certain influenza strains. If there are too many mutations, however, then a virus may lose its ability to replicate, an observation used in development of treatments for *polio* and *hepatitis C* which exacerbate respective virus mutations [59, 60]. VIDO thus provides the term:

*disordered virus* = def Acellular self-replicating organic structure having some clinically abnormal arrangement of viral components (e.g. viral capsid, viral DNA/RNA).

Viruses falling in this class may be associated with diseases much different from those associated with viruses that do not exhibit disorder. Terms for viral components are imported to VIDO: from GO, *viral nucleocapsid, viral capsid, capsomere, viral envelope*; from the Chemical Entities of Biological Interest (ChEBI) ontology (https://bioportal.bioontology.org/ontologies/CHEBI) [61, 62], *nucleic acid* and *ribonucleic acid*; from the Protein Ontology (https://bioportal.bioontology.org/ontologies/PR), *protein* and *viral protein*.

**Infectious structure.** The term "pathogen" is indexed to species or to stages in the developmental cycle of a species. A given virus may engage in mutual symbiosis with one species, while exhibiting pathogenic behavior towards others [63, 64]. Mature plants are often susceptible to different pathogens than developing plants [65–68]. We capture virus pathogenicity in VIDO in steps. From IDO Core [16], we import dispositions borne by pathogens and infectious agents, as follows:

*pathogenic disposition* = def Disposition borne by a material entity to establish localization in, or produce toxins that can be transmitted to, an organism or acellular structure, either of which may form disorder in the entity or immunocompetent members of the entity's species.

*infectious disposition* = def Pathogenic disposition borne by a pathogen to be transmitted to a host and then become part of an infection in that host or in immunocompetent members of the same species as the host.

The class *infectious agent* in IDO Core is a subclass of *organism*, and so cannot include instances of *virus*. To address this issue, the term *infectious structure* was developed to parallel the IDO Core term *infectious agent* and to provide a logically defined subclass of *acellular self-replicating organic structure*. The term *infectious disposition* bridges infectious acellular structures and infectious organisms since instances of each bear an infectious disposition. Moreover, the logical definitions of *infectious structure* and *infectious agent* are such that, though the former is a defined subclass of *acellular self-replicating organic structure* and the latter a subclass of *organism*, they are both inferred subclasses of *pathogen*.

As discussed in [16], establishment of localization used in *pathogenic disposition* is characterized using the IDO Core term *establishment of localization in host* representing tethering or adhesion to a host, while "formation of disorder" abbreviates *appearance of disorder*, which is a *process* that *results in formation of* a *disorder*. The definition of *pathogenic disposition* is meant to reflect a temporal ordering between establishment of localization and appearance of disorder. This is reflected explicitly in the logical axioms associated with the class. Similarly, in the definition of *infectious disposition* there is an intended temporal ordering between transmission to a host–represented by *pathogen transmission process* imported from the Pathogen Transmission Ontology (https://bioportal.bioontology.org/ontologies/PTRANS)–and becoming part of an infection–represented by the IDO Core *process of establishing an infection*. A pathogen bearing an *infectious disposition* that generates disorder in a host will have been transmitted to the host prior to establishing localization in the host and will have established an infection prior to the appearance of disorder.

The complexity of the definitions of *pathogenic disposition* and *infectious disposition* reflect the variety of pathogen examples documented in contemporary literature. Consider *S. aureus*, an opportunistic pathogen [56] in humans. We count *S. aureus* as a pathogen, even when it does not realize disorder in a host, since it is nevertheless disposed to localize in a human host and generate disorder if given the opportunity. This is a BFO disposition of *S. aureus* as it is an "internally-grounded" property of the entity [32]. That is, it is part of the material basis of *S. aureus* to generate disorder in human hosts if given the chance. This is analogous to the way salt has a disposition to dissolve, based on its lattice structure, independently of whether it ever realizes this disposition. Opportunistic pathogens are not pathogens because of an opportunity; they are pathogens because they are disposed to localize and cause disorder in a host.

Consider now, *C. botulinum*, a pathogen which produces a toxin and which may produce a spore ingested by humans. This bacterium is a pathogen for adult humans since the toxins often result in disorder when ingested. Furthermore, *C. botulinum* may cause infection in human infants if, say honey colonized by *C. botulinum* is ingested. The sugar content of honey inhibits *C. botulinum* growth, but in the low-oxygen, low-acid intestines of human infants, spores can localize, grow, and produce toxins resulting in disorder. Thus, *C. botulinum* counts as a human infant pathogen. Nevertheless, because *C. botulinum* is not itself disposed to invade or be transmitted to human infants, we do not say the bacterium is infectious [69]. Being part of an infection is not itself sufficient for something to be counted as infectious. Pathogens bearing an *infectious disposition* must be disposed to both transmit and become part of an infection. Many opportunistic pathogens, for example, are not infectious.

Consider lastly, the respective definitions of *infectious disposition* and *pathogenic disposition* address instances where mutations in hosts may block realization of disorder or infection. In such cases, an infectious pathogen may nevertheless be transmissible and cause disorder or infection in others. For example, HIV-1 is a pathogen that may localize in a host with CCR-5 mutations [70] that block the virus from attaching to host cells, and so block pathogenesis to AIDS. Similarly, *P. falciparum* may be transmitted to a host with a sickle-cell trait that blocks manifestation of the disease malaria [71, 72]. However, *P. falciparum* and HIV-1 count as pathogens even if they do not result in the formation of disorders for hosts with a sickle-cell trait or CCR-5 mutation, respectively. IDO Core reflects this characterization. Each pathogen may be transmitted to immunocompetent members of the same species as the host, and so count as bearing instances of *infectious disposition* and *pathogenic disposition*. Note, the fact that *P. falciparum* and HIV-1 do not result in the formation of disorders in hosts with sickle-cell traits or CCR-5 mutations should not suggest that there are no clinical abnormalities associated with these traits or mutations. Individuals with, say, CCR-5 mutations do exhibit clinical abnormalities, and so do exhibit disorders. But these disorders are due to the CCR-5 mutation rather than the HIV-1 infection.

Whenever an *infectious disposition* is realized, this is always in some site in a host, via some transmission to that host, and some generation of infection and disorder in that host. *Infectious structures*–such as viruses–bear this disposition. For example, each SARS-CoV-2 virus is disposed to be transmitted to hosts, localize, cause infection, and result in disorder.

**Pathogen host.** Until recently, microbiologists, immunologists, virologists, and others studying pathogenesis have engaged in either host-centered or pathogen-centered pathogenesis research [73–77]. Each approach has led to impressive research results. But emphasizing one aspect of host-pathogen interactions at the expense of the other may leave valuable questions unanswered [78]. Emphasis, for example, solely on pathogenic factors of SARS-CoV-2 will provide only a partial explanation of various pathogenesis pathways observed in clinical settings. IDO Core and VIDO prioritize neither host nor pathogen in representation of pathogens and associated diseases, adopting the Damage Response Framework (DRF) for guidance in development of relevant terms [79–82]. According to the DRF, pathogenesis results from interactions between both host and pathogen interacting primarily through host damage, which is a function of the intensity and degree of host response and pathogen factors. Host and pathogen interactions thus influence manifestations of signs, symptoms, and disease. IDO Core now defines hosts and pathogens in terms of roles and allows that acellular structures may also be pathogen hosts, such as when a virus hosts a virophage:

*host role* = def Role borne by either an organism whose extended organism contains a distinct material entity, or an acellular structure containing a distinct material entity, realized in use of that structure or organism as a site of reproduction or replication.

*pathogen host role* = def Host role borne by an organism or acellular structure having a pathogen as part.

Following BFO, roles are "externally grounded" realizable entities that may be gained or lost based on circumstance without necessarily involving material change to their bearer, such as the role a student acquired once enrolled in a university.

The Symptom Ontology (https://bioportal.bioontology.org/ontologies/SYMP) provides a extensive terminological content for representing symptoms owing to viral infection, such as *fever*, *taste alteration*, and so on [83]. Given the importance of asymptomatic carriers in viral infection spread, moreover, attention is also given in IDO Core to:

*symptomatic carrier role* = def Pathogen host role borne by an organism whose extended organism contains a pathogen bearing an infectious disposition towards the host, and the host has manifested symptoms of the infectious disease caused by the pathogen.

*asymptomatic carrier role* = def Pathogen host role borne by an organism whose extended organism contains a pathogen bearing an infectious disposition towards the host, and the host has no symptoms of the infectious disease caused by the pathogen.

*subclinical infection* = def Infection that is part of an asymptomatic carrier.

The definition of the term *subclinical infection* reflects standard use of the terms "subclinical" and "asymptomatic" while allowing for asymptomatic clinically abnormal infections. VIDO extends *subclinical infection* to *subclinical virus infection*, namely, those subclinical infections caused by a virus. These remarks bring us full circle to the term *disorder* introduced above, since clinical abnormality is associated with disorder. When that disorder stems from infection it counts as an:

*infectious disorder* = def Disorder that is part of an extended organism which has an infectious pathogen part, that exists as a result of a process of formation of disorder initiated by the infectious pathogen.

And when the adverted pathogen is a virus, it falls in the VIDO class:

*virus disorder* = def Infectious disorder that exists as a result of a process of formation of disorder initiated by a virus.

**Viral disease.**   Medical researchers draw a distinction between symptoms and signs, a distinction which OBO Foundry ontologies respect (from OGMS [38]):

*symptom* = def Process experienced by the patient which can only be experienced by the patient, that is hypothesized to be clinically relevant.

*qualitative sign* = def Abnormal observable quality of a part of a patient that is hypothesized to be clinically relevant.

*processual sign* = def Abnormal processual entity occurring in a patient that is hypothesized to be clinically relevant.

An asymptomatic carrier infected with SARS-CoV-2 likely exhibits signs indicating that the infection is clinically abnormal, such as ground-glass opacities. Such asymptomatic carriers exhibit an instance of the VIDO class *virus disorder* which is the material basis of a *viral disease*:

*infectious disease* = def Disease whose physical basis is an infectious disorder.

*viral disease* = def Infectious disease inhering in a virus disorder that is a disorder due to the presence of the virus.

Worth noting is that these definitions are consistent with the CDC's case criteria definitions adopted between April 5, 2020 and February 28, 2023, which indicate that the presence of the SARS-CoV-2 genome or relevant antigens in an individual is sufficient to count as a case of COVID-19, asymptomatic or not [84–86]. A *viral disease* may be realized in a disease course:

*infectious disease course* = def Disease course that is the realization of an infectious disease.

*viral disease course* = def Infectious disease course whose physical basis is a virus disorder that is clinically abnormal in virtue of the presence of the relevant virus.

Here *infectious disease* and *infectious disease course* are imported from IDO Core, and are subclasses, respectively, of *disease* and *disease course*, which are imported from OGMS.

**Viral epidemiology.**   Changes in viral disease and infection incidence are among the targets of epidemiological investigation. VIDO imports from IDO Core:

*infectious disease incidence* = def Quality that inheres in an organism population and is the number of realizations of an infectious disease for which the infectious disease course begins during a specified period.

*infectious disease incidence rate* = def Quality that inheres in an organism population and is the infectious disease incidence proportion per unit time.

*infectious disease incidence proportion* = def Quality that inheres in an organism population and is the proportion of members of the population not experiencing an infectious disease

course at the beginning of a specified period and in whom the infectious disease begins during the specified period.

*organism population* = def Aggregate of organisms of the same species.

Additionally, VIDO imports from IDO Core other important epidemiological terms, such as *infection prevalence*, *infectivity*, and *infectious disease mortality rate*. Each are specifically dependent entities inhering in some material entity, though not always in some organism population. For example, *infectivity* is a quality inhering in instances of *pathogen*. Additionally, VIDO imports from IDO Core:

*infection incidence* = def Quality that inheres in an organism population and is the number of organisms in the population that become infected with a pathogen during a specified period.

on which infectious disease incidences depend, as infectious disease realizations require infection.

A quality process profile is a type of process which tracks changes of specific qualities in material entities over time [26]. For example, a patient's temperature will likely fluctuate over time, as will many other qualities of the patient. The specific fluctuations of temperature in the patient over time is a *process profile*, which reflects common abstractions used in clinical diagnosis and testing and manifesting in charts prepared from time-series data. Changes in qualities of clinical interest may follow several patterns, each of which can be defined as a subclass of *process profile*. A patient's temperature may exhibit a linear increase followed by a linear decrease. Similarly, there are *process profile* instances of cyclical patterns, for instance the seasonal patterns of *influenza* [87]. Such patterns can be tracked in VIDO by classes such as:

*viral disease incidence profile* = def Infectious disease incidence profile comprised of a series of determinate viral disease incidence qualities caused by a specific virus in a population over time.

*viral disease incidence proportion profile* = def Infectious disease incidence proportion profile comprised of a series of viral disease incidence proportion qualities caused by a specific virus per unit time.

*viral disease incidence rate profile* = def Infectious disease incidence rate profile comprised of a series of viral disease rate qualities caused by a specific virus per unit time.

## Extending VIDO to CIDO

VIDO serves as a bridge between IDO Core and the IDO extension ontologies representing specific viral diseases. Of particular importance during the pandemic has been the Coronavirus Infectious Disease Ontology (CIDO; https://bioportal.bioontology.org/ontologies/CIDO), developed by the He Group at the University of Michigan. CIDO provides terminological content that facilitates representations of coronavirus genome, protein structures, epidemiological surveillance, vaccine development, and treatment options. The ontology has been used to annotate data pertaining to 136 known anti-coronavirus drugs [6], as well as in the identification of approximately 110 candidate drugs [7] for potential drug repurposing projects with respect to COVID-19 [88]. More recently, CIDO has been employed as a general framework for understanding host-pathogen interactions [78]. IDO Core, VIDO, and CIDO development teams work together closely in the interest of ensuring ontology alignment.

CIDO will extend from VIDO by adopting, among other terms:

*coronavirus disease* = def Viral disease inhering in a coronavirus disorder.

*coronavirus disease course* = def Viral disease course that is the realization of some coronavirus disease and has as a participant a coronavirus.

This extension example illustrates a downward population recipe useful for aligning CIDO terms to VIDO, by starting with a given virus term from the latter, and restricting subclasses based on features of coronaviruses and associated diseases. Moreover, common coronavirus features can be reused from OBO ontologies to complement the CIDO characterization of the virus, such as the viral envelop glycoprotein spikes [89, 90]. Some such terms are specializations of terms from the Protein Ontology (https://bioportal.bioontology.org/ontologies/PRO), e.g. *spike glycoprotein (SARS- CoV-2)*. CIDO covers existing and novel coronaviruses in general, and so provides resources for detailed comparison of coronavirus biological profiles. Fig 4 illustrates various links between VIDO and CIDO, and the IDO Core and GO ontologies.

**SARS-CoV-2 pathogenesis.**  Characterizing pathogenesis to COVID-19 is aided by terms such as:

*COVID-19* = def Coronavirus disease inhering in a SARS-CoV-2 disorder.

*COVID-19 disease course* = def Coronavirus disease course that is the realization of some COVID-19 disease and has participant SARS-CoV-2.

Ontologically precise representation of COVID-19 pathogenesis is crucial for understanding the range of symptoms and signs which appear across demographics [91–94]. Ontological representation of COVID-19 pathogenesis is aided by reusing OBO Foundry ontology terms, resulting in the following definition:

*SARS-CoV-2 pathogenesis* = def Coronavirus pathogenesis process realization of an infectious

disposition inhering in SARS-CoV-2 or a SARS-CoV-2 population, having at least the proper process parts:

1. pathogen transmission,

2. establishment of localization in host,

3. process of establishing an infection, and

4. appearance of a virus disorder.

Instances of *SARS-CoV-2 pathogenesis* are asserted as part of some *COVID-19 disease course*. The term *coronavirus pathogenesis* will ultimately be imported to CIDO, as a subclass of the VIDO term *viral pathogenesis*, itself a subclass of:

*pathogenesis* = def Process that generates the ability of a pathogen to induce disorder in an organism.

which is imported from GO. As defined, pathogenesis is a success term [25], in that it encompasses formation of disorder in an entity. Of course, this is not meant to imply that SARS-CoV-2 infections necessarily lead to successful pathogenesis. SARS-CoV-2 infections, for example, may not lead to host disorder, in which case there would be no pathogenesis. Just as important as it is to represent SARS-CoV-2 pathogenesis to COVID-19, adequate representation of the target domain requires representation of pathogenesis to *acute respiratory distress syndrome* (ARDS), one of the leading causes of death in those infected by SARS-CoV-2 [95, 96]:

*acute respiratory distress syndrome* = def Progressive and life-threatening pulmonary distress in the absence of an underlying pulmonary condition, usually following major trauma or surgery.

which may be imported from the Experimental Factor Ontology (https://bioportal. bioontology.org/ontologies/EFO). Similar remarks apply to other diseases associated with SARS-CoV-2 pathogenesis.

SARS-CoV-2 pathogenesis involves transmission of SARS-CoV-2 virions. From PTRANS (https://bioportal.bioontology.org/ontologies/PTRANS) is imported:

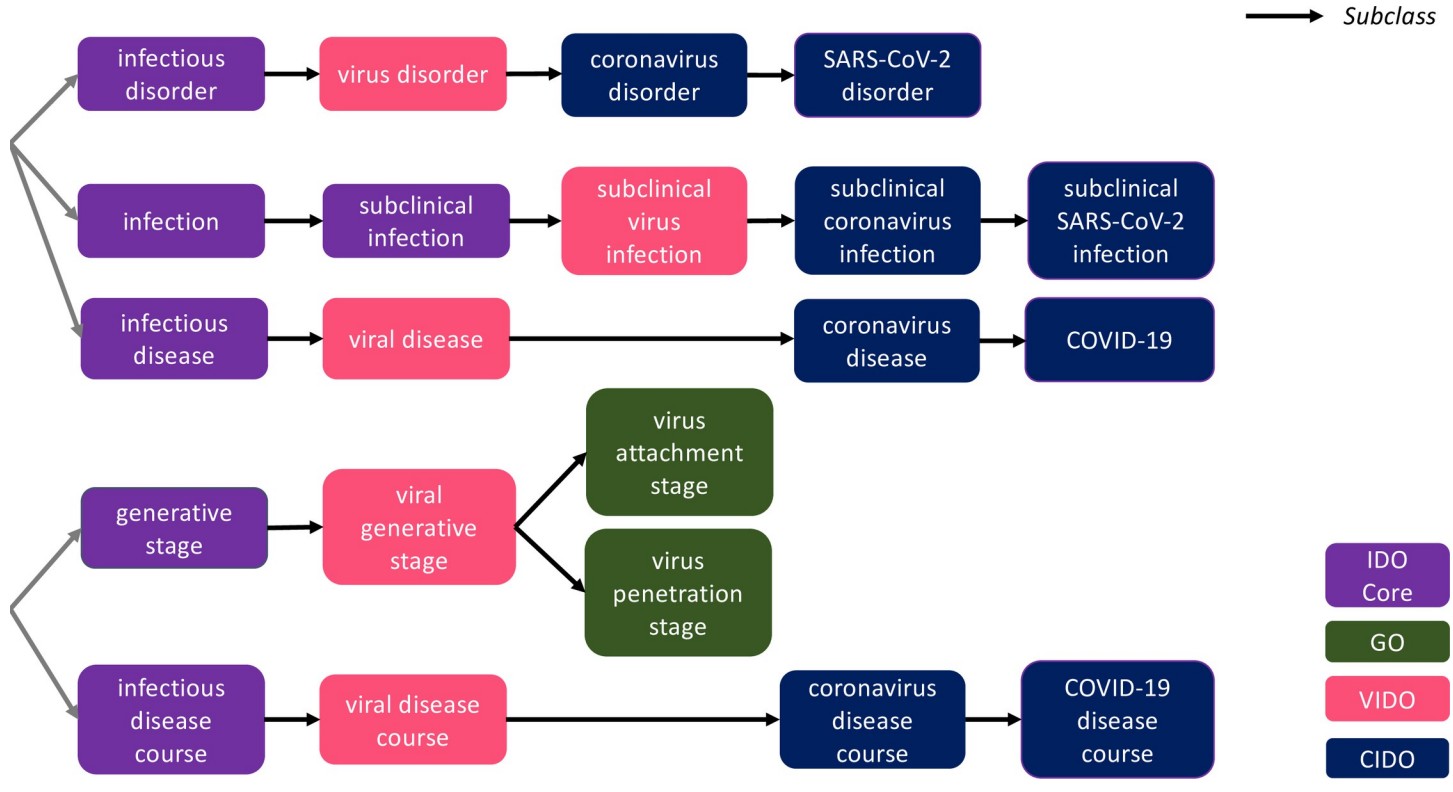

**Fig 4. Relationships among GO, IDO Core, VIDO, and CIDO.**

*pathogen transmission process* = def Process during which a pathogen is transmitted directly or indirectly to a new host.

From which SARS-CoV-2 specific terms can be constructed. Additionally, IDO Core provides important role terms relevant to pathogen transmission, such as:

*pathogen transporter role* = def Role borne by a material entity in or on which a pathogen is located, from which the pathogen may be transmitted to a new host.

An important subclass *fomite role*–roughly, a pathogen transporter role borne by a non-living entity–may feature in SARS-CoV-2 transmission via instances of *fomite role* bearing:

*respiratory droplet* = def Respiratory secretion composed of a bounded portion of liquid which maintains its shape due to surface tension.

*respiratory droplet SARS-CoV-2 fomite* = def Respiratory droplet fomite with SARS-CoV-2 part.

Knowledge of transmission steps supports strategies designed to break the transmission chain.

Worth noting is that the OBO library ontology APOLLO-SV (https://bioportal.bioontology.org/ontologies/APOLLO-SV) also contains terms, such as *contact tracing* and *quarantine control strategy*, which may be leveraged to represent virus-specific transmission control strategies.

**SARS-CoV-2 replication.**   SARS-CoV-2 pathogenesis involves replication in a host. The term *virus replication* is defined in VIDO as a subclass of the IDO Core term *replication*, specifically:

*virus replication* = def Replication process in which a virus containing some portion of genetic material inherited from a parent virus is replicated.

And instances of *viral disease course* and *virus pathogenesis* have *virus replication* as parts. SARS-CoV-2 replication occurs within an:

*incubation process* = def Process beginning with the establishment of an infection in a host and ending with the onset of symptoms by the host, during which pathogens are multiplying in the host.

Which occupies an *incubation interval* and may precede a *communicability interval*. The corresponding process during which SARS-CoV-2 hosts bear a contagiousness disposition has proper part some *latency process* which itself has an *eclipse process* as part::

*communicability interval* = def One-dimensional temporal region during which a pathogen host bears a contagiousness disposition.

*latency process* = def Process beginning with the establishing of an infection in a host and ending when the host becomes contagious, during which pathogens are multiplying in the host.

*eclipse process* = def Process beginning with the establishment of a virus in a host and ending with the first appearance of a virion following viral release, during which an infecting virus is uncoating to begin genome replication.

The last are specific to viruses, and so specific to VIDO. Viral dormancy is a virus-specific term from VIDO occurring over a:

*viral dormancy interval* = def One-dimensional temporal region on which a virus is no longer replicating but remains within a host cell and which may be reactivated to begin replication again.

Viral dormancy is characteristic of familiar viruses such as *varicella zoster* and *herpes simplex*.

VIDO includes as a temporal subdivision of a virus developmental process:

*virus generative stage* = def Infectious structure generative stage that is a temporal subdivision of a virus developmental process.

Subclasses of which include the stages through which viruses may proceed during replication:

*virus attachment stage* = def Virus generative stage during which a virion protein binds to molecules on the host surface or host cell surface projection.

*virus penetration stage* = def Virus generative stage during which a virion or viral nucleic acid breaches the barriers of a host.

*SARS-CoV-2 attachment stage* = def Virus attachment stage during which SARS-CoV-2 bonds with a host cell.

*SARS-CoV-2 penetration stage* = def Virus penetration stage during which SARS-CoV-2 penetrates a host cell.

**SARS-CoV-2 susceptibility.**   Only cells with certain features are susceptible to SARS-CoV-2 infection [16]. For example, successful infection in humans typically involves SARS-CoV-2 attachment to alveolar epithelial cells through angiotensin-converting enzyme 2 (ACE2) receptors [97–99]. Cells lacking ACE2 receptors seem protected from attachment by SARS-CoV-2. Those with receptors can be represented in CIDO using:

*SARS-CoV-2 adhesion susceptible cell* = def Virus adhesion susceptible cell with a functional receptor part bearing an adhesion disposition realized in a SARS-CoV-2 attachment stage.

*adhesion disposition* = def Disposition borne by a macromolecule that is the disposition to participate in an adhesion process.

Where *adhesion disposition* is imported from IDO Core and *virus adhesion susceptible cell* defined in VIDO. The ACE2 functional receptor is defined in the Protein Ontology (https://bioportal.bioontology.org/ontologies/PRO):

*angiotensin-converting enzyme 2* = def A protein that is a translation product of the human ACE2 gene or a 1:1 ortholog thereof.

Attachment is frequently followed by cell penetration, where cell cleavage is aided by transmembrane protease serine 2 (TMPRSS2) prior to SARS-CoV-2 cell membrane fusion [100, 101]. These observations motivate introducing terminological content for defining *SARS-CoV-2 penetration susceptible cell* such as:

*SARS-CoV-2 penetration disposition* = def Virus penetration disposition borne by a functional receptor complex that is the disposition to participate in a SARS-CoV-2 penetration process.

Ontological representation of the SARS-CoV-2 replication cycle provides targets for disruption or regulation of that cycle, which is important to rational drug design [102–106]:

*negative regulation of SARS-CoV-2 attachment* = def Negative regulation of coronavirus replication process that stops, prevents, or reduces the frequency of some SARS-CoV-2 attachment stage.

*negative regulation of SARS-CoV-2 penetration* = def Negative regulation of coronavirus replication that stops, prevents, or reduces the frequency of some SARS-CoV-2 penetration stage.

Following our strategy of linking VIDO and CIDO, parent classes of negative regulation of coronavirus classes have a proper home in CIDO, while their parent classes–negative regulation of viruses more generally–have a proper home in VIDO.

**Annotations.** Coverage in VIDO and CIDO can be illustrated by annotation of coronavirus research articles.

Consider the following overview of SARS-CoV-2 pathogenesis (compare bold with Fig 5): Following **replication**, **cell lysis** of **SARS-CoV-2 coronavirus virions** causes **host cells** to **release molecules** which **function** to warn nearby **cells**. When **recognized** by **epithelial cells**, **endothelial**, and **alveolar macrophages**, **proteins** such as **IL-6**, **IP-10**, and **MCPI**, are **released** which **attract T cells**, **macrophages**, and **monocytes to** the **site of infection**, **promoting inflammation**. In **disordered immune systems**, **immune cells accumulate in** the **lungs**, then **propagate to** and **damage** other **organs**. In **normal immune systems**, **inflammation attracts T cells** which **neutralize** the **virus at** the **site of infection**. **Antibodies circulate**, preventing **SARS-CoV-2 infection**, and **alveolar macrophages recognize SARS-CoV-2** and **eliminate virions** via **phagocytosis** [92, 107, 108].

In a more ontologically oriented language, we speak of the relevant part of a host's immune response as being disposed to manifest a response that eliminates SARS-CoV-2 infection, while SARS-CoV-2 has a disposition to block manifestation of this immune system response. Consider next a color-coded selection from the *Lancet* [109] concerning SARS-CoV-2:

"The **viral load**s **in throat swab**s and **sputum sample**s peaked **at** around **5–6 days after symptom onset**, ranging from around **$10^4$ to $10^7$ copies per mL during** this **time**."

SARS-CoV-2 infected hosts contain the highest concentration of SARS-CoV-2 virions–the *viral load*–during the *incubation interval* [110]. Viral load is a common measurement of the proportion of virions to fluid, and for SARS-CoV-2 is frequently measured from host sputum. VIDO provides the resources for annotating virus quantification:

*viral load* = def Quality inhering in a portion of fluid that is the proportion of virions to volume of that portion of fluid

Our color-coding of the above passage from the *Lancet* models term reuse across existing ontologies. For example, developers can use VIDO and CIDO terms alongside terms from the Common Core Ontology (https://github/com/CommonCoreOntology/CommonCoreOntologies) such as *is measured by*, *measurement information content entity*, *has integer value*, *uses measurement unit*, and *milliliter measurement unit*. Other virus

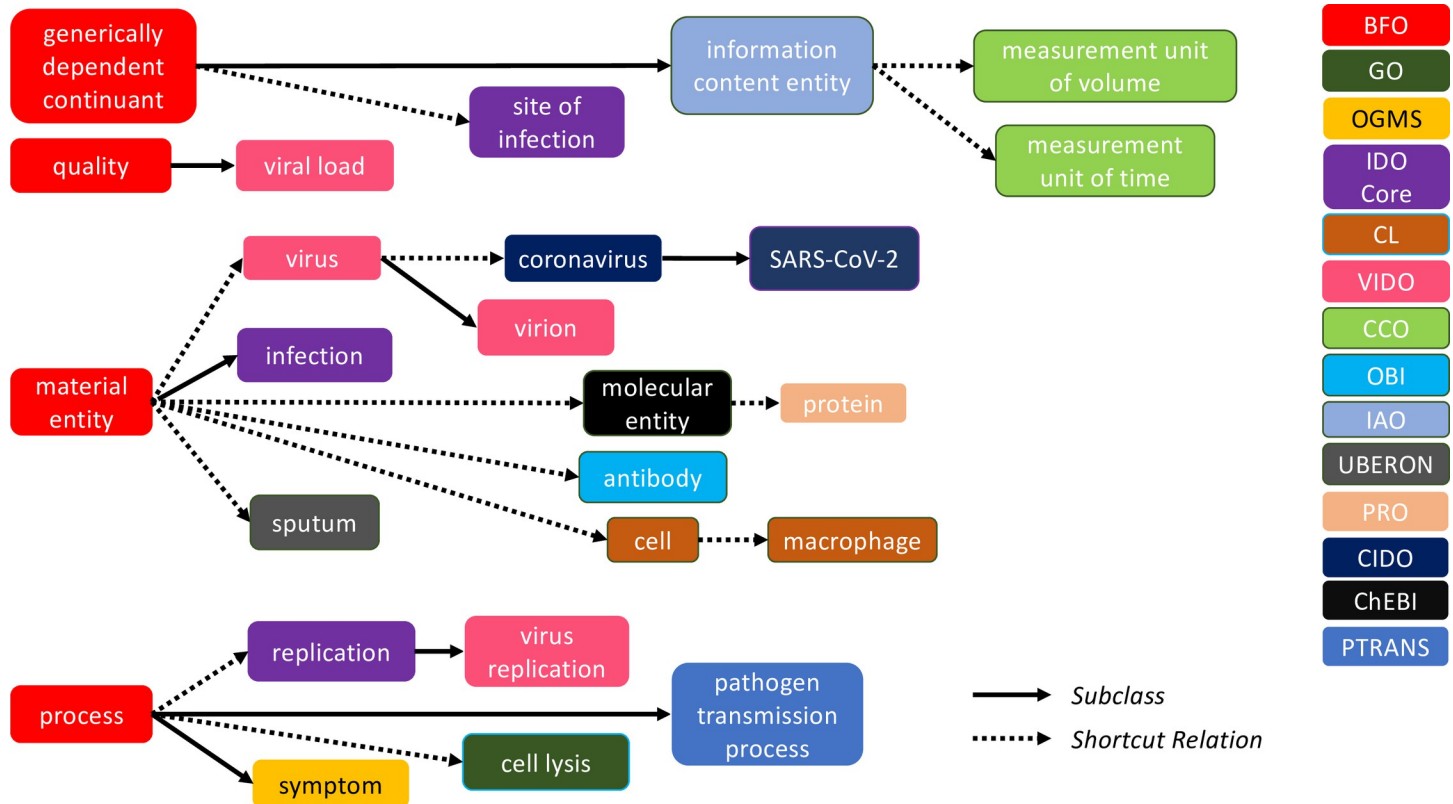

**Fig 5. Relationships among BFO, GO, OGMS, IDO Core, the Cell Ontology (CL), VIDO, the Common Core Ontology (CCO), OBI, IAO, the Uber-Anatomy Ontology (UBERON), CIDO, ChEBI, PRO (Protein Ontology), and PTRANS.**

quantification metric terms, such as *multiplicity of viral infection* - the ratio of virions to susceptible cells in a target area–can be found in VIDO as well.

## Discussion

Motivated to standardize virus ontology extensions of IDO Core, we have developed VIDO and provided a recipe for connecting VIDO to more virus-specific extensions of IDO Core, illustrated by connecting VIDO to CIDO. Summarizing our results, we have introduced *acellular structure* as a parent class to *virus*, motivated using the Baltimore classification to model viruses rather than the International Committee on Taxonomy of Viruses classification, revised IDO Core's pathogen and host classes to accommodate acellular structures, and extended IDO Core's *infectious disease*, *infectious disease course*, and infectious disease epidemiology classes to cover viruses. We then introduced bridge classes in VIDO to better align IDO Core and CIDO, illustrating throughout how CIDO terminological content can be extended and enriched to represent SARS-CoV-2 pathogenesis, associated transmission processes, virus transporters, replication stages and associated temporal extents, as well as pathogenesis regulation. Our attention was then turned to annotations of texts concerning SARS-CoV-2 and COVID-19, by which we highlighted how ontologies using common vocabularies may be seamlessly interwoven to provide broad annotation coverage for the domain.

VIDO and CIDO are not the only ontologies developed to support curation of COVID-19 data [111–114]. However, most alternatives are stand-alone initiatives, and so subject to the silo problems typically found in ontologies developed outside the scope of the OBO Foundry

and with no attention to its principles. That said, VIDO and CIDO developers have participated in harmonization efforts aimed at semantic integration across COVID-19 ontologies [115]. Notably, harmonization efforts have resulted in the deprecation of the COVID-19 Infectious Disease Ontology (IDO-COVID-19)–introduced in a preprint version of the current paper [116]–with parties agreeing its scope was subsumed by CIDO. Additionally, VIDO and CIDO have been used to highlight ontology conflict resolution strategies [117]. It is not uncommon for ontology researchers working independently in nearby domains to construct overlapping ontology content.

The harmonization efforts of the VIDO and CIDO development teams signal to the wider ontology community our willingness to reuse terms where possible and obsolete terms or cede terms to other ontologies when needed.

VIDO and CIDO enable extensive representation of virus-related research. The very scope of VIDO provides challenges, however, as does the specificity of CIDO. For these reasons, attempts have been made to foster community-driven development of both. The development team for each ontology spanned disciplines in the life sciences, and to ensure the computational viability of the formal representation of each ontology, included specialists in logic. Often, terms were developed then presented to domain specialists for vetting, after which they were refined through discussion.

As in the case of all scientific ontologies, refinement will continue as research advances, and further collaborators are welcome. Interested parties may contact the corresponding author to be invited to on-going VIDO development meetings and may contact co-author He for invitation to development meetings concerning CIDO. Additionally, collaborators are encouraged to raise issues on respective GitHub issue trackers for VIDO (https://github.com/infectious-disease-ontology-extensions/VIDO) and CIDO (https://github.com/CIDO-ontology/cido).

The existence of IDO Core extensions covering infectious disease-causing entities other than viruses suggests a need for the creation of reference ontology extensions of IDO covering bacteria, fungi, and parasites. To that end, development of the Bacteria Infectious Disease Ontology is underway (https://github.com/infectious-disease-ontology-extensions/Bacteria-Infectious-Disease-Ontology) as is the development of the Fungal Infectious Disease Ontology (https://github.com/SydCo99/MIDO). The methodology illustrated in the development of VIDO provides a recipe for such reference ontology creation. Additionally, the methodology illustrated in the development of CIDO provides a recipe for the creation of novel virus-specific ontologies, namely, by extending them from existing virus ontologies. Adoption of these methodologies by developers during ontology construction will significantly reduce the labor involved in ontology creation. Related, linking research on infectious diseases to developments on non-infectious diseases is no less important than our focus here. In this respect, VIDO and CIDO benefit from alignment with IDO Core, which itself aligns with the Ontology of General Medical Science (OGMS), whose scope extends beyond infectious disease. What this means in practice is that, for example, kidney disease [118] and cancer [119] researchers accurately using the OGMS vocabulary to represent data, invariably use ontology terms and methodologies common to IDO Core, VIDO, and CIDO, thereby lowering barriers to data integration and interoperability.

VIDO and CIDO are being used to annotate host-coronavirus protein-protein interactions, in the interest of developing more effective treatment strategies for those infected by SARS-CoV-2 or variants [37]. While various treatments have been authorized for emergency use [120], there is significant room for improvement. Rather than focus on a single drug to treat infected patients, VIDO and CIDO developers have pursued investigating drug cocktail strategies to improve treatment outcomes. Foundational to these investigations has been proper

characterization of viral proteins playing different roles in host-coronavirus interactions which impact pathogenesis [78].

From another direction, VIDO and CIDO have been used in automated electronic health-record annotation [121], in particular those involving COVID-19 data, which highlights the importance of providing researchers with terminological content relevant to nearby domains. Recent developments at the intersection of ontology engineering and machine learning research have, moreover, motivated the need for formally well-defined ontologies in machine learning pipelines using minimal data [122]. By exploiting formal axioms in, for example, the Gene Ontology, impressive zero-shot predictions for protein functions can be generated [123]. We believe the formal axiomatization of VIDO makes it a particularly promising ontology for inclusion in zero-shot research and intend to explore how VIDO may supplement machine learning efforts in future work.

## Conclusion

Ontologies provide important tools for overcoming contemporary big data challenges. It is incumbent on working ontologists representing life science research to seek harmonization with nearby ontologies, else we run the risk of reinstating the same big data challenges ontologies have previously been so successful at addressing. VIDO represents a substantial effort to characterize viruses in general, in a collaborative, computationally tractable manner. CIDO too represents a significant effort to characterize coronaviruses in a specific, no less collaborative, no less computationally tractable manner. Connecting VIDO to CIDO improves semantic interoperability among IDO Core-conformant infectious disease ontologies and, moreover, improves interoperability with other BFO-conformant ontologies, ranging from the OBO Foundry to numerous other ontology projects employing BFO as a top-level architecture. Consequently, our work provides researchers resources for gathering and coordinating life science data while avoiding issues that so frequently undermine automating integration and analyses of the data flood in which we so often find ourselves [124–126].

## Acknowledgments

Many thanks to Asiyah Yu Lin for assistance in VIDO development and harmonization; to Darren Natale and Sydney Cohen for helpful critical feedback on earlier drafts; to Amanda Hicks and Neil Otte for comments on the VIDO rdf files; to participants at the 2022 International Conference on Biomedical Ontologies, with particular thanks to Alexander Diehl and Chris Stoeckert for their helpful feedback before, during, and after the conference. Figures were designed by or in consultation with Rain Yuan.

## Author Contributions

**Conceptualization:** John Beverley, Shane Babcock, Gustavo Carvalho, Lindsay G. Cowell, Sebastian Duesing, Yongqun He, Regina Hurley, Eric Merrell, Richard H. Scheuermann, Barry Smith.

**Formal analysis:** John Beverley, Lindsay G. Cowell, Sebastian Duesing, Yongqun He, Barry Smith.

**Investigation:** John Beverley, Shane Babcock, Gustavo Carvalho, Lindsay G. Cowell, Yongqun He, Regina Hurley, Eric Merrell, Barry Smith.

**Methodology:** John Beverley, Gustavo Carvalho, Regina Hurley, Eric Merrell, Richard H. Scheuermann, Barry Smith.

**Project administration:** Gustavo Carvalho.

**Supervision:** John Beverley, Barry Smith.

**Validation:** Shane Babcock, Gustavo Carvalho, Eric Merrell, Richard H. Scheuermann.

**Writing – original draft:** John Beverley.

**Writing – review & editing:** John Beverley, Shane Babcock, Gustavo Carvalho, Lindsay G. Cowell, Sebastian Duesing, Yongqun He, Regina Hurley, Eric Merrell, Richard H. Scheuermann, Barry Smith.

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
