## [Decision Letter · Decision Letter 0]

3 Feb 2023

PONE-D-22-32364

Coordinating Virus Research: The Virus Infectious Disease Ontology

PLOS ONE

Dear Dr. Beverley,

Thank you for submitting your manuscript to PLOS ONE. After careful consideration, we feel that it has merit but does not fully meet PLOS ONE’s publication criteria as it currently stands. Therefore, we invite you to submit a revised version of the manuscript that addresses the points raised during the review process.

We look forward to receiving your revised manuscript.

Kind regards,

Barry L. Bentley, Ph.D.

Academic Editor

PLOS ONE

Journal Requirements:

- https://pubmed.ncbi.nlm.nih.gov/34275487/?

In your revision ensure you cite all your sources (including your own works), and quote or rephrase any duplicated text outside the methods section. Further consideration is dependent on these concerns being addressed.

“NO”

“NO”

Reviewers' comments:

Reviewer's Responses to Questions

**Comments to the Author**

1. Is the manuscript technically sound, and do the data support the conclusions?

Reviewer #1: Yes

Reviewer #2: Yes

2. Has the statistical analysis been performed appropriately and rigorously? 

Reviewer #1: N/A

Reviewer #2: Yes

3. Have the authors made all data underlying the findings in their manuscript fully available?

Reviewer #1: Yes

Reviewer #2: Yes

4. Is the manuscript presented in an intelligible fashion and written in standard English?

Reviewer #1: Yes

Reviewer #2: Yes

5. Review Comments to the Author

Reviewer #1: This manuscript describes VIDO as an extension of IDO. It is well written and easy to understand. There are a few minor issues to be addressed.

on page 2, the first instance of "data silos" is italicized. It is not clear why it is, but it would be nice to explain what is meant by the phrase. In certain circles, it is well understood, but the audience of plos one is broad enough that it would be nice to have a short explanation.

On pages 14 and 15, the definitions for host role and pathogen role seem to be reversed. The host is not meant to be acellular?

On page 18, coronaviris disease and disease course terms and mentioned, yet no coronavirus terms could be found in VIDO. Has this work not been completed yet? If so, please state this or say when it will be released.

On page 22, the sentence that starts with "With participants found from familiar..." near the top of the page is an incomplete sentence.

On page 22 at the bottom of the page, the link to a human ACE2protein is mentioned. I would have expected a link to a higher node species independent protein. Is VIDO human specific? Many other species can be infected with coronaviruses. Is that data outside the scope of VIDO? If so, please mention this.

On page 25, the fact that collaborators are welcomed could use an explanation - are there weekly calls that others can join or should it be via github requests or just by email or all of the above?

On page 25, the Fungal and Bacterial Infectious Disease Ontologies are welcomed news, however, how and if IDO and its children relate to non-infectious diseases would be very interesting to projects that must represent all types of diseases, including those that are not infectious.

If you have any data on usage or projects that are currently collaborating, that information would be nice to have.

Extra information that would be very interesting to address, but may be out of scope of this article, so are not required for publication:

Any discussion regarding the interoperability of IDO and VIDO with other ontologies and any plans to migrate IDO terms to other ontologies having the same or similar terms, for example 'host' and 'viral latency' are in GO would be nice to have. 'Infectious disease' is in many other ontologies, 'syndrome' is on OGMS, 'latency' is in PATO. 'Adhesion disposition' and 'incubation process' are not virus specific terms, so should live outside of VIDO, for example.

Reviewer #2: Some specific examples of data silos rather than just mentioning might be helpful in the article to entertain the readers. In the same vein, just mentioning of the machine learning algorithm with the reference[6] to predict efficient Covid vaccine demands more specific examples of machine learning projects where the ontologies might play a crucial role (as claimed in the article). Having said that, perhaps, there is a need to tightly couple or link the article’s developmental ontologies to any machine learning projects in characterizing viruses in a collaborative, computationally tractable manner, more importantly, for the researchers across the disciplines.

A more clarification is needed on how incorporating the Baltimore classification provides developers in VIDO more specific virus classes within a succinct, navigable, and ontological structure as claimed in the article.

In the discussion section that follows the results, perhaps a paragraph of a high-level summary of the results section could be fruitful which can then be subsequently linked with the points discussed on the VIDO and CIDO.

Finally, perhaps, a conclusions section might be useful to deliver a powerful statement in connecting the VIDO to the CIDO.

6. PLOS authors have the option to publish the peer review history of their article (what does this mean?). If published, this will include your full peer review and any attached files.

Reviewer #1: No

Reviewer #2: No

---

## [Author Response · Author response to Decision Letter 0]

17 Mar 2023

Response to Reviewers

We very much appreciate the reviewers’ comments throughout; we feel our submission has been improved by working through the helpful feedback offered.

Reviewer #1

This manuscript describes VIDO as an extension of IDO. It is well written and easy to understand. There are a few minor issues to be addressed.

on page 2, the first instance of "data silos" is italicized. It is not clear why it is, but it would be nice to explain what is meant by the phrase. In certain circles, it is well understood, but the audience of plos one is broad enough that it would be nice to have a short explanation.

 We appreciate the reviewer’s feedback and have added a brief description of data silos as well as motivating examples. We have removed the italics since they appear to have been distracting. 

On pages 14 and 15, the definitions for host role and pathogen role seem to be reversed. The host is not meant to be acellular?

 We have revised the definitions for these terms in the interest of clarity. The term “host” in IDO is intended to be broad enough to cover organism hosts and acellular hosts, since there are scenarios in which viruses and may host other virus or bacterial pathogens, e.g. virophages. We appreciate the reviewer’s comment; this was a helpful clarification. 

On page 18, coronavirus disease and disease course terms and mentioned, yet no coronavirus terms could be found in VIDO. Has this work not been completed yet? If so, please state this or say when it will be released.

 Because “coronavirus” and related terms concern a specific virus, they are better suited for inclusion in ontologies that extend from VIDO. In this case, the Coronavirus Infectious Disease Ontology would be the appropriate home for coronavirus terms, and indeed includes many. VIDO and CIDO development teams are working to align existing CIDO terms to VIDO, as evidenced by our harmonization effort. 

On page 22, the sentence that starts with "With participants found from familiar..." near the top of the page is an incomplete sentence.

 Thank you; we have rephrased this sentence. 

On page 22 at the bottom of the page, the link to a human ACE2 protein is mentioned. I would have expected a link to a higher node species independent protein. Is VIDO human specific? Many other species can be infected with coronaviruses. Is that data outside the scope of VIDO? If so, please mention this.

 Though we did not intend readers to draw this conclusion, on reflection, we understand why the reviewer did. VIDO is intended to cover infectious viral diseases for any host. Similarly, CIDO is intended to cover infectious coronavirus diseases for any host. We have revised this discussion to clarify that our example is not intended to suggest VIDO is human-specific.

On page 25, the fact that collaborators are welcomed could use an explanation - are there weekly calls that others can join or should it be via github requests or just by email or all of the above?

 Excellent feedback, we agree more should be added. The VIDO team meets monthly and the CIDO team meets semi-monthly. We are always happy to have collaborators join. We have clarified how interested parties may become involved in the projects. We have also added reference to the respective github pages, where issues may be raised and pull requests open for suggested changes. 

On page 25, the Fungal and Bacterial Infectious Disease Ontologies are welcomed news, however, how and if IDO and its children relate to non-infectious diseases would be very interesting to projects that must represent all types of diseases, including those that are not infectious. If you have any data on usage or projects that are currently collaborating, that information would be nice to have.

 We agree on the relevance of terms in IDO and its extensions to non-infectious disease research. IDO itself extends from the Ontology of General Medical Science (OGMS) which provides resources to characterize disease of more than just the infectious variety. In our work revising IDO, we make a point to distinguish pathogens like c. botulinum from infectious pathogens, in pursuit of sharpening our focus on the latter in IDO. Nevertheless, the scope of OGMS provides resources for representing data on non-infectious diseases. We have commented on this in the discussion section and added relevant references. 

Extra information that would be very interesting to address, but may be out of scope of this article, so are not required for publication:

Any discussion regarding the interoperability of IDO and VIDO with other ontologies and any plans to migrate IDO terms to other ontologies having the same or similar terms, for example 'host' and 'viral latency' are in GO would be nice to have. 'Infectious disease' is in many other ontologies, 'syndrome' is on OGMS, 'latency' is in PATO. 'Adhesion disposition' and 'incubation process' are not virus specific terms, so should live outside of VIDO, for example.

 Our revision of the discuss section was motivated in no small part by the reviewer’s helpful comments. Here we include responses in some detail:

• IDO was developed alongside members of the GO community and so alignment was and remains a priority. To our knowledge, GO does not include a general term for “latency process” but does include terms for more specific, e.g. viral latency, processes. Ideally, GO’s term would extend from the more general IDO term. That said, it is not our intention to introduce a “viral latency” term to VIDO, since such a term already exists in GO. We instead introduce in the paper a “viral latency interval” which is intended to be the temporal extent of that process. Even more ideally, GO would cede “viral latency” to VIDO, along with numerous other virus-specific terms. That said, as the reviewer’s point suggests, there are other terms overlapping between GO and VIDO. Greater alignment between these two ontology resources – as well as between GO and CIDO - is work we intend to pursue going forward.

• GO has many terms that use “host” but we were unable to identify a term in GO intended as broadly as “host” in IDO Core. The closest we found in GO was “host cellular component” which is asserted to be a synonym to “host organism” but which has a definition indicating it is intended to concern cell components of organisms. In any event, IDO developers intend “host” to be broad enough to cover GO use of the term, though not conversely. In that way, GO host-specific terms may extend from the more general IDO Core host term. 

• We are happy to include “syndrome” from OGMS where there is a need stemming from viral infection. In the event of such a need, we would use the OGMS syndrome term as the parent to a virus-specific syndrome term. 

• “Adhesion disposition” and “incubation process” are IDO Core terms, though it is understandable why this may have been unclear in the text. We have revised the text to make this clearer. 

• PATO is a more challenging case. On the one hand, there is a note on PATO “latency” indicating it is up for obsoletion, so alignment may be moot. On the other hand, if it is not obsoleted, we worry that including this in VIDO will conflict directly with our top-level hierarchy BFO. This is because in PATO “latency” is a subclass ultimately of BFO’s “specifically dependent continuant” class and “latency” is defined as a “quality of a process”. However, in BFO processes cannot bear qualities, as discussed here: 

Smith B. (2012) Classifying Processes: An Essay in Applied Ontology. Ratio. (4):463-488. 

For these reasons we are inclined to work towards having PATO developers revise their use of “latency” to align with IDO and VIDO. 

 We again stress how helpful the reviewer’s comments have been as we revise this document. We hope to have addressed the concerns raised. 

Reviewer #2 

Some specific examples of data silos rather than just mentioning might be helpful in the article to entertain the readers. 

 Thank you for the clarificatory feedback; it is much appreciated. We have provided a brief description of data silos in the introduction, alongside examples of how they occur. 

In the same vein, just mentioning of the machine learning algorithm with the reference[6] to predict efficient Covid vaccine demands more specific examples of machine learning projects where the ontologies might play a crucial role (as claimed in the article). Having said that, perhaps, there is a need to tightly couple or link the article’s developmental ontologies to any machine learning projects in characterizing viruses in a collaborative, computationally tractable manner, more importantly, for the researchers across the disciplines.

 We have highlighted uses of VIDO and CIDO in data mining and data enrichment pipelines and added recent research providing optimism for the use of ontologies in machine learning pipelines, in particular, for zero-shot goals. We have removed the paragraph concerning applications of VIDO and CIDO to the creation of a gold corpus standard, as this project is no longer being pursued. 

A more clarification is needed on how incorporating the Baltimore classification provides developers in VIDO more specific virus classes within a succinct, navigable, and ontological structure as claimed in the article.

 We have revised the document to make clear that we are contrasting the benefits of using the Baltimore classification against employing a traditional Linnean hierarchy, as many infectious disease ontologists have in the past. We have included a figure to illustrate how unwieldy such hierarchies can be. 

In the discussion section that follows the results, perhaps a paragraph of a high-level summary of the results section could be fruitful which can then be subsequently linked with the points discussed on the VIDO and CIDO.

 Excellent advice, thank you. We have incorporated your suggestion.

Finally, perhaps, a conclusions section might be useful to deliver a powerful statement in connecting the VIDO to the CIDO.

 Excellent advice, thank you. We have incorporated your suggestion.

---

## [Editor Report · Decision Letter 1]

17 Apr 2023

Coordinating Virus Research: The Virus Infectious Disease Ontology

PONE-D-22-32364R1

Dear Dr. Beverley,

We’re pleased to inform you that your manuscript has been judged scientifically suitable for publication and will be formally accepted for publication once it meets all outstanding technical requirements.

Kind regards,

Barry L. Bentley, Ph.D.

Academic Editor

PLOS ONE
---

## [Editor Report · Acceptance letter]

22 Jun 2023

PONE-D-22-32364R1 

Coordinating Virus Research: The Virus Infectious Disease Ontology 

Dear Dr. Beverley:

I'm pleased to inform you that your manuscript has been deemed suitable for publication in PLOS ONE. Congratulations! Your manuscript is now with our production department. 

Kind regards, 

on behalf of

Dr Barry L. Bentley 

Academic Editor

PLOS ONE